# Drosophila CG17003/leaky (lky) is required for microtubule acetylation in early germ cells in Drosophila ovary

**Matthew Antel, Taylor Simao, Muhammed Burak Bener, Mayu Inaba** *

Department of Cell Biology, University of Connecticut Health, Farmington, CT, United States of America

* inaba@uchc.edu

## Abstract

Microtubule acetylation is found in populations of stable, long-lived microtubules, occurring on the conserved lysine 40 (K40) residue of α-tubulin by α-tubulin acetyltransferases (αTATs). α-tubulin K40 acetylation has been shown to stabilize microtubules via enhancing microtubule resilience against mechanical stress. Here we show that a previously uncharacterized αTAT, Drosophila CG17003/leaky (lky), is required for α-tubulin K40 acetylation in early germ cells in Drosophila ovary. We found that loss of lky resulted in a progressive egg chamber fusion phenotype accompanied with mislocalization of germline-specific Vasa protein in somatic follicle cells. The same phenotype was observed upon replacement of endogenous α-tubulin84B with non-acetylatable α-tubulin84B$^{K40A}$, suggesting α-tubulin K40 acetylation is responsible for the phenotype. Chemical disturbance of microtubules by Colcemid treatment resulted in a mislocalization of Vasa in follicle cells within a short period of time (~30 min), suggesting that the observed mislocalization is likely caused by direct leakage of cellular contents between germline and follicle cells. Taken together, this study provides a new function of α-tubulin acetylation in maintaining the cellular identity possibly by preventing the leakage of tissue-specific gene products between juxtaposing distinct cell types.

## Introduction

Microtubules play essential roles on various cellular processes including polarized trafficking, mitosis, migration and determining cellular rigidity [1]. Microtubule dynamics are regulated by post-translational modifications (PTMs) to the α- and β- tubulin subunits. One such PTM is α-tubulin acetylation, which, when occurring on the conserved lysine 40 (K40) residue of α-tubulin, has been reported to be enriched in populations of stable microtubules [2–4]. The enzymes responsible for acetylation of α-tubulin are α-tubulin acetyltransferases (αTATs), identified first in C. elegans as Mec-17 [5, 6] and are highly conserved across species [7, 8]. α-tubulin K40 acetylation (α-K40 acetylation, here after) enhances the resilience of microtubules against mechanical stresses [9–11]. Mutant studies in various organisms, including C. elegans [12], zebrafish [6], mice [13, 14] and Drosophila [15], revealed that α-K40-acetylation is

**Data Availability Statement:** All relevant data are within the manuscript and its Supporting Information files.

**Funding:** This research is supported by R35GM128678 from National Institute for General

Medical Sciences and start-up funds from UConn Health (to M.I.). The funders had no role in study design, data collection and analysis, decision to publish, or preparation of the manuscript.

**Competing interests:** The authors declare no competing interests.

commonly required in mechanosensory neurons, likely through maintaining cellular rigidity to resist external forces. Importantly, the specific requirement of α-K40-acetylation in mechanosensory neurons implies that α-K40-acetylated microtubules may have a qualitatively distinct role from other populations of microtubules. Despite the universality of microtubule function and occurrence of α-K40 acetylation in any cell types, the requirement of α-K40-acetylation beyond the neuronal system has been poorly understood.

*Drosophila* oogenesis proceeds in structures called ovarioles, where egg chambers at multiple stages of development are aligned in a spatiotemporally ordered manner. The most anterior unit of the ovariole is the germarium, where germline stem cells (GSCs) continuously produce differentiating germ cells. At the tip of germarium, GSCs divide asymmetrically to produce a cystoblast, which divides 4 more times and forms a 16-cell syncytium or a cyst (Fig 1A) [16–18]. One of the 16 germ cells within the cyst becomes the oocyte, while the other 15 cells become nurse cells, which support oocyte development [16–18]. Somatic follicle cells (FCs) are derived from stem cells residing in the germarium [19–21], which proliferate to give rise to epithelial layer and encapsulate rapidly enlarging germline cysts [16, 22]. This process repeatedly occurs with precise spatial and temporal resolution, providing a perfect model system to study how germline and soma interact and coordinate with each other to complete successful oogenesis [16, 23].

Microtubules play diverse roles throughout oogenesis, such as oocyte determination and differentiation [24, 25]. In early stages of oogenesis, microtubules associate with a germ cell-specific endoplasmic reticulum-like organelle called the fusome, known to be a major microtubule organizing center (MTOC) of the cyst (Fig 1A) [26]. The fusome branches through dividing germ cells within the cyst [27, 28]. The microtubules emanating from this MTOC are polarized in a manner that directs vesicular trafficking [29–31]. In later stages of oogenesis, microtubules are involved in the transport of materials into the maturing oocyte from nurse cells [32–39]. Microtubules also play more general roles in the ovary, such as establishing and maintaining cell-cell junctions (reviewed in [40]) and contributing to the determination of cellular rigidity in cooperation with the actomyosin cytoskeleton [41, 42].

In the ovary, acetylated α-tubulin has been reported to be enriched on fusomes in the dividing germ cells of early oogenesis [43]. The role of α-K40 acetylation in oocyte development has been unknown. In this study, we show that a previously uncharacterized αTAT, *Drosophila CG17003/leaky* (*lky*), is required for α-tubulin K40 acetylation in early germ cells in *Drosophila* ovary and demonstrates an unexpected roles of α-K40 acetylated microtubules in cell-fate maintenance during oogenesis.

## Results

### Lky localizes to fusomes and is responsible for α-K40-acetylation in the early germline

*CG17003* is predicted to be an αTAT based on sequence homology [15]. Henceforth, we refer to *CG17003* as *leaky* (*lky*). We found that Lky-GFP expressed by *nosGal4*, a germline specific driver, localizes at the fusome (Fig 1B and 1C), which has been reported to be enriched for acetylated microtubules [43]. Using fluorescence *in situ* hybridization (FISH), we detected *lky* expression in the early stage of the germline, especially in region 1-2a of germ cells in which fusomes are present (Fig 1D), suggesting that Lky may function specifically in fusomes of early germ cells.

To examine whether Lky is responsible for α-K40-acetylation at the fusome, We generated a knockout fly in which the entire *lky* coding region is deleted (*lky*^KO) (see Methods, S1A and S1B Fig). *lky*^KO homozygous flies were viable and fertile and *lky*^KO ovaries appeared normal in

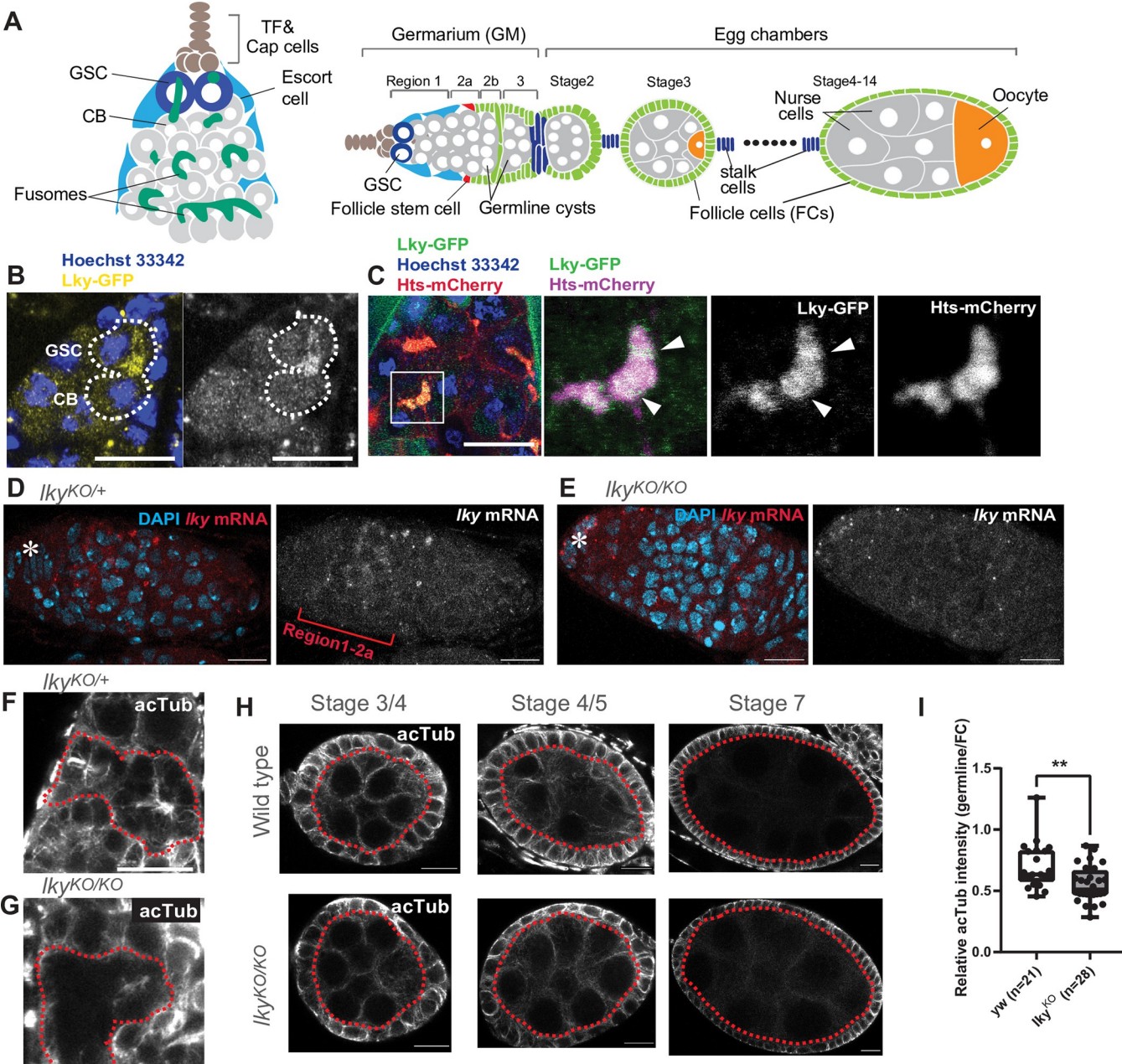

**Fig 1. Lky localizes to fusomes and is responsible for α-K40-acetylation in early germline. A**) (Left) A schematic showing Region 1-2a germarium (GM), the anterior-most unit of the ovariole. TF = terminal filament, GSC = germline stem cell, CB = cystoblast. TF and Cap cells forms the stem cell niche. A GSC (magenta) continuously divides to produce the differentiating daughter, the cystoblast (CB). The CB enters 4 incomplete mitotic division to form a 16-cell cyst. Fusomes (green) connect dividing germ cells in the cyst. (Right) A schematic of morphologically distinguishable regions/stages of an ovariole. Follicle stem cells (red) localize between region 2a and 2b in the germarium and proliferate to produce follicle cells or FCs (light green). FCs encapsulate developing germline cysts and a population of FCs is specified as stalk cells (blue). **B**) A representative image of a GSC-preCB pair (encircled by a white dotted line) expressing Lky-GFP (yellow) under the germline-specific driver, *nosGal4*. Hoechst 33342 (blue) makes nuclei. **C**) A representative image of Lky-GFP in a 8-cell cyst. Inset shows fusome with Lky-GFP accumulated around ring canals (or mid-body rings, indicated by white arrowheads). Right three panels in **C** show magnified images of the inset. Note; because Lky-GFP signal in the fusome reduces after fixation, live tissue was used for imaging for **B** and **C**. **D, E**) Representative images of FISH against *lky* mRNA in *lky*^KO/+ germaria (**D**) or *lky*^KO homozygous germaria (**E**). *lky* transcript was detected in the early stage of germline, Region 1-2a, whereas it was completely absent in *lky*^KO ovary. **F, G**) Representative images of immunofluorescent (IF) staining of germaria for anti-acetylated-αTubulin (acTub) from heterozygous (control) (**F**) or homozygous (**G**) *lky*^KO. Region 2a cysts are encircled by red broken lines. **H**) Representative images of germline cysts of egg chambers at the indicated stage stained with acTub. **I**) A graph shows relative intensities of acTub staining in wild type (yw) and *lky*^KO germline cysts up to stage 5. Cytoplasmic germline measurements were divided by the cytoplasmic acTub fluorescence intensity of adjacent follicle cells used for internal control. Box plot shows 25–75% (box), median (band inside) and minimum to maximum (whiskers) with each data point representing a single

measurement. Adjusted P value from Student's t-test is provided. For **B-I**, flies of days 0 to 7 post-eclosion were used. For all data points, a minimum of 20 ovaries were used from two individual crosses. Scale Bars; 10 μm. Asterisks indicate approximate niche location.

young animals (0–7 days post-eclosion). The *lky* mRNA FISH signal was completely absent in *lky*^KO^ ovaries, confirming a loss of *lky* expression in the *lky*^KO^ (Fig 1D and 1E). Consistent with a previous report [43], we observed acetylated microtubules enriched in the cell boundary of early syncytium cysts (2–16 cell cysts) where the fusome branches (Fig 1F). In *lky*^KO^ ovaries, levels of acetylated α-tubulin (acTub) were reduced or absent in these cysts (Fig 1G), and significant reduction was also observed in early germline cysts (up to stage 5) (Fig 1H and 1I). We did not detect any noticeable changes of acTub staining in the somatic cells of *lky*^KO^ ovaries (Fig 1H). These results indicate that Lky is responsible for α-K40-acetylation specifically in early stages of germline cysts, and later stages (beyond stage 6) and somatic cells likely use other αTATs for α-K40-acetylation.

Since we still observed acTub staining in somatic cells and other stages of germline, we examined if the other known *Drosophila* αTAT, *dTAT* (*CG3967*) [15] is responsible for α-K40-acetylation in the cells in which Lky's function is absent (S1C–S1G Fig). Germ cells in *dTAT*^KO^ germaria maintained acetylated fusome microtubules (S1E Fig), indicating that the germ cells mainly depend on Lky but not on dTAT for α-K40-acetylation. In contrast, we observed loss of acTub in somatic cells of *dTAT*^KO^ ovaries (S1E Fig). The *dTAT* and *lky* double mutant (*Double KO*, S1F Fig) showed the anterior region of the germarium almost completely absent for acTub. These observations suggest a role for dTAT in early somatic cells and a role for Lky in the germline. However, we still observed acTub in the posterior of germaria and later staged egg chambers in *Double KO* ovaries (S1F and S1G Fig), indicating the existence of a third αTAT responsible for α-K40-acetylation in later stages.

## α-K40-acetylation by Lky is required for maintenance of oogenesis during fly aging

Oogenesis proceeds in units called ovarioles, where egg chambers consisting of a germline cyst surrounded by a follicular epithelium are aligned in a spatiotemporal order. Each egg chamber is separated from the next by a string of somatic stalk cells (Fig 1A) [44–48].

Although *lky*^KO^ ovariole structure appeared normal in young animals (0–7 days after eclosion), we often observed incomplete separation of egg chambers in older *lky*^KO^ ovaries. The phenotype worsened as the animal aged (Fig 2A–2G). In germarium, we frequently observed fused cysts, where 2 or more cysts were continuous. Later stage egg-chambers also show incomplete separation. For example, a single FC layer was encapsulating 30 nurse cells and 2 oocytes in *lky*^KO^ while wildtype egg chambers contain only one oocyte (Fig 2C–2F).

Expression of *lky* using a germline-specific driver (*nosGal4*), but not a somatic cell driver (*tjGal4*), rescued the egg chamber fusion phenotype of *lky*^KO^ ovaries, and the knockdown of *lky* in the germline, but not in the somatic cells, showed the egg chamber fusion phenotype (Fig 2H and 2I). These data indicate that Lky is required in the germarium for normal egg chamber separation.

To test if the egg chamber fusion phenotype is caused by defective α-K40-acetylation, we attempted to rescue the aberrant egg chambers with expression of an α-tubulin K40Q transgene (*UASp-αTub*^K40Q^) in which lysine (K) 40 of α-tubulin84B (the most abundant α-tubulin isoform in *Drosophila* [49]) is mutated to glutamine (Q). The α-tubulin K40Q mutation has been shown to mimic acetylated α-tubulin [6, 50, 51]. Using *nosGal4* and *tjGal4*, we drove cell type-specific expression of αTub^K40Q^ in the *lky*^KO^ background. We found that germline-

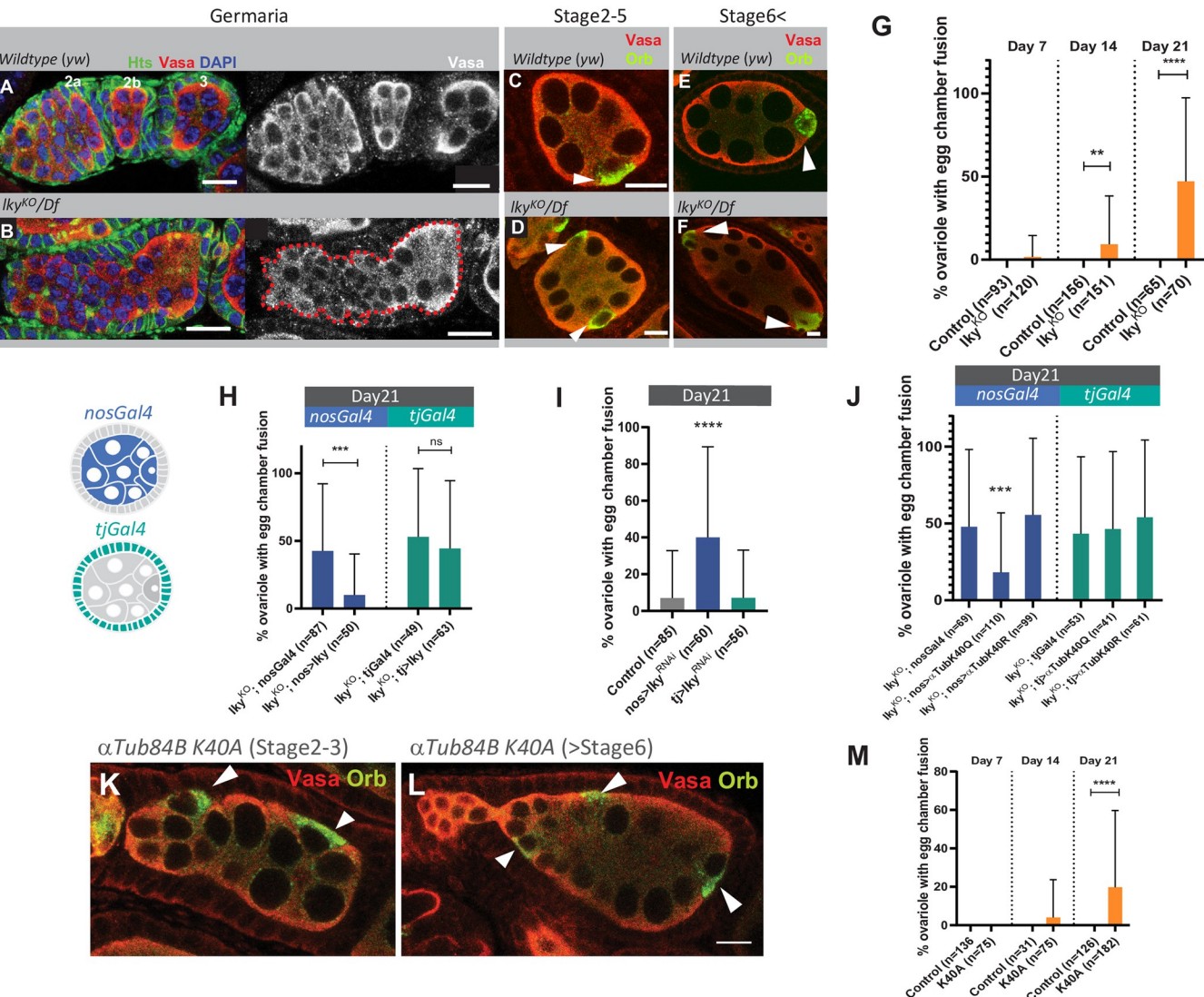

**Fig 2. α-K40-acetylation by Lky is required for maintenance of oogenesis during fly aging. A-F)** Representative images for comparison of *wildtype* (*yw*, **A, C, E**) and *lky*^KO (*lky*^KO/*Df*, **B, D, F**) ovarioles, showing the egg chamber fusion phenotype in *lky*^KO. White arrowheads in **C-F** indicate oocytes (Orb, green) in indicated stages of egg chambers of wildtype (**C, E**) and *lky*^KO (**D, F**). **G**) A graph shows frequency of ovarioles with egg chamber fusion, scored at 7, 14, and 21 days post-eclosion for the indicated genotypes. **H**) A graph shows frequency of ovarioles with egg chamber fusion in *lky*^KO with or without germline (*nosGal4* driven)- or somatic cell (*tjGal4* driven)- specific expression of *lky*. **I**) A graph shows frequency of ovarioles with egg chamber fusion in ovaries of *lky* RNAi induced under the germline (*nosGal4*)- or somatic cell (*tjGal4*) drivers. **J**) A graph shows frequency of ovarioles with egg chamber fusion after germline (*nosGal4* driven)- or somatic cell (*tjGal4* driven)- specific expression of *αTubulin*^K40Q or *αTubulin*^K40R transgenes. Flies were used at day 21 post-eclosion for **H-J**. **K-L**) Representative images of *αTub84B*^K40A homozygous knock-in ovarioles showing egg chamber fusion at indicated stages. Flies were used at day 21 post-eclosion. **M**) A graph shows frequency of ovarioles with egg chamber fusion in *αTub84B*^K40A knock-in ovaries at indicated ages. Heterozygous flies (*αTub84B*^K40A/+) were used for the control. Adjusted P values from Šidák's multiple comparisons test are shown for G, H, and J, and Dunnett's multiple comparisons test for I. For all data points, a minimum of 20 ovaries were used from two individual crosses. Scale Bars: 10 μm.

specific (but not somatic cell-specific) expression of αTub^K40Q rescued the egg chamber fusion in *lky*^KO ovaries (Fig 2J). In contrast, a non-acetylatable α-tubulin84B K40R (αTub^K40R), which has lysine (K) 40 mutated to arginine (R), was not able to rescue the aberrant egg chamber defect (Fig 2J). Moreover, mutant flies in which the lysine (K) 40 residue of the endogenous *α-tubulin84B* gene is replaced to alanine (A) (αTub^K40A, which mimics acetylation-null α-tubulin [52]) exhibited a similar phenotype to *lky*^KO flies, often showing fused egg chambers

which worsened with age (Fig 2K–2M). These data indicate that the egg chamber fusion observed in *lky^{KO}* is due to an impairment of α-K40-acetylation.

Why does the egg chamber fusion caused by loss of Lky only appear in aged ovaries? We reasoned there must be three possible explanations. First, Lky is expressed only in aged tissues and young tissue may use other αTATs. We found this is unlikely as *lky* depletion showed loss of α-K40-acetylation even in young ovaries (Fig 1F–1I). Moreover, we did not detect an age-dependent change in *lky* mRNA level (S2A Fig). The second explanation could be that age-dependent changes of tissues may sensitize tissues to the loss of α-K40-acetylation. Third, loss of α-K40-acetylation may require significant time for the phenotype to appear. To distinguish the second and third possibilities, we induced *lky* RNAi expression by heat shock (see *Methods* for details) either in young (at day 1 post-eclosion) or aged (at day 21 post-eclosion) animals and compared the outcomes. Interestingly, the egg chamber fusion phenotype developed almost immediately (3 days) after RNAi induction in aged animals, but did not develop in young animals (S2B–S2D Fig). These results support the second possibility, in which age-dependent physiological changes sensitize tissues to the loss of α-K40-acetylation.

## Vasa protein is often detected in FCs in *lky^{KO}* ovaries

Other than egg-chamber fusion, we noticed that germaria or egg chambers of *lky^{KO}* ovaries contained FCs that stained positive for Vasa, a germline-specific protein, as well as the FC marker Traffic Jam (TJ) (Fig 3A–3F). Vasa-positive FCs were typically observed in any stage of egg chambers, often forming a cluster in FC layer (Fig 3F). The frequency of ovarioles containing FCs positive for Vasa progressively increased with age (Fig 3G). Vasa mislocalization was significantly rescued by germline-specific (but not somatic cell-specific) expression of *lky* (Fig 3H), suggesting that loss of Lky function in the germline causes this phenotype.

Vasa-positive FCs were also observed in the homozygous *αTub^{K40A}* knock-in line, in which α-tubulin84B is considered acetylation-null (Fig 3I–3K) [52], indicating that the loss of α-K40-acetylation directly results in the mislocalization of germline components in FCs.

These data suggest that α-K40-acetylation in the germline is required for preventing the mislocalization of germline-specific proteins in FCs.

## Short-term Colcemid treatment recapitulates Vasa mislocalization observed in *lky^{KO}* ovaries

How does loss of α-K40-acetylation result in mislocalization of germline contents in FCs? We sought two possibilities: first, that a germline without α-K40-acetylation alters FC fate, which leads to expression of germline genes in FCs, and second, that loss of α-K40-acetylation leads to disruption of the physical barrier between the germline and FCs resulting in "leakage" of germline cytoplasmic contents into neighboring FCs. To distinguish these possibilities, we incubated egg chambers in the presence of the microtubule depolymerizing drug Colcemid for a short period of time. This treatment should result in immediate loss of microtubules. We found that the treatment of wildtype ovaries with Colcemid for 30 minutes was sufficient to delocalize massive amounts of Vasa in FCs, resembling the pattern of Vasa-positive FCs observed in *lky^{KO}* (Fig 4A–4C). Staining Colcemid-treated ovaries for Arm, a marker of adherens junctions, revealed that leakage of Vasa into somatic cells was associated with disruption of adherens junctions on the germ cell-FC boundary (Fig 4D and 4E), suggesting that the junction between germ cells and the FC boundary may be weakened upon loss of microtubules. Moreover, visualizing plasma membrane by FM4-64 membrane dye revealed the absence of plasma membrane on the germ cell-FC boundary (Fig 4G).

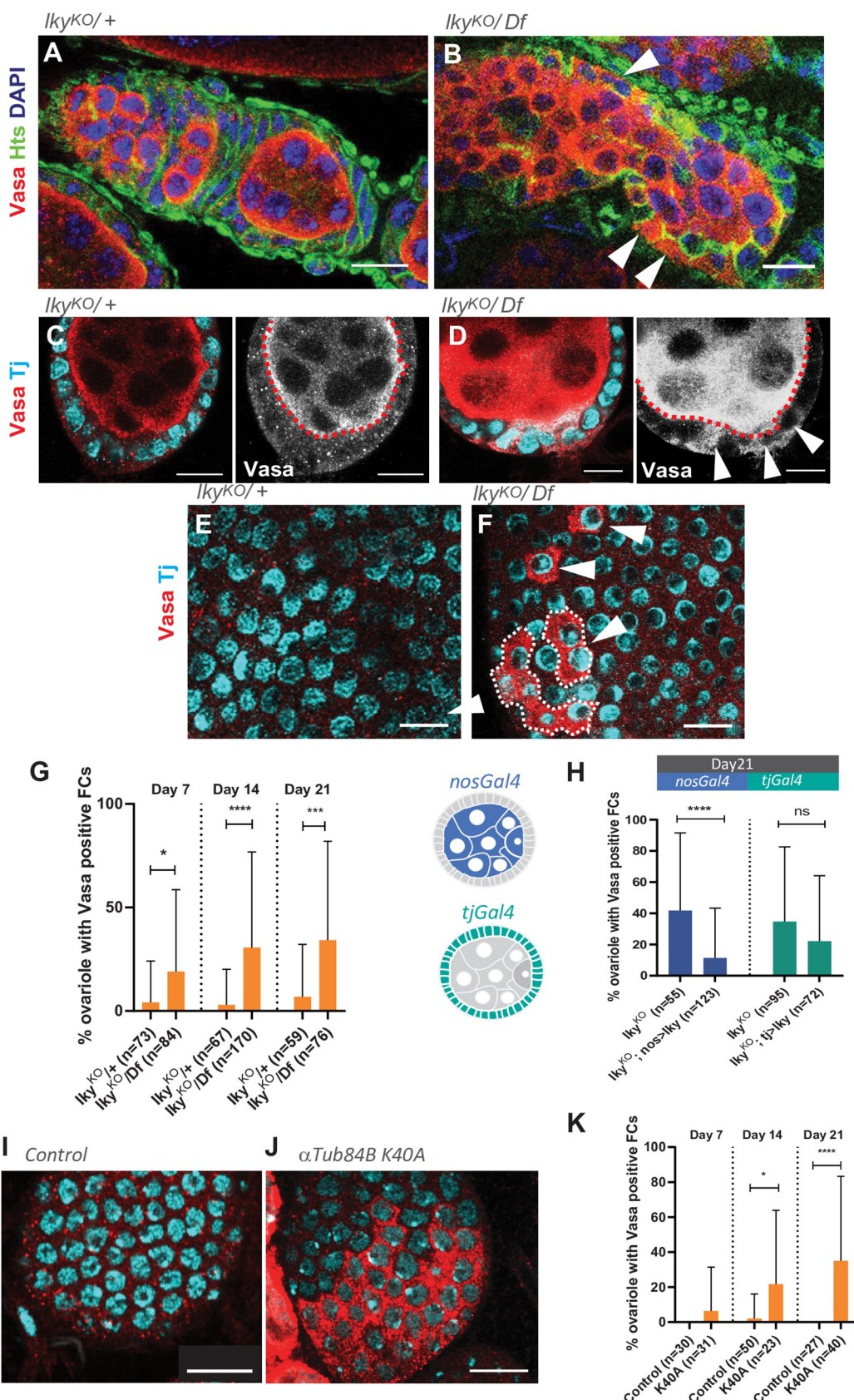

**Fig 3. Vasa protein is often detected in FCs in *lky^KO* ovaries. A, B**) Representative images of IF staining of germaria. *lky^KO*/ *Df* image shows abnormal localization of Vasa in the FC layer (white arrowheads). **C-F**) IF staining of Stage3-4 egg chambers for Vasa and TJ, a FC marker, showing Vasa protein in TJ-positive FCs (white arrowhead) in *lky^KO*. **C** and **D** shows surface views of an egg chamber, **E** and **F** show cross sections. For **A** to **F**, flies were used at days 7–14 post-eclosion. **G**) A frequency of ovarioles with any ectopic Vasa localization detected in FCs, comparing control versus *lky^KO* for 7, 14, and 21 days. Percentages of ovarioles containing any FCs which stain positive for Vasa (Vasa in FC cytoplasm) were manually counted. **H**) A graph shows frequency of ovarioles containing Vasa-positive FCs in indicated genotypes. Germline-specific expression of *lky* in the germline (under the *nosGal4* driver) reduced the frequency, but somatic cell-specific expression of *lky* (under the *tjGal4* driver) had no effect. **I, J**) Representative surface views of Vasa-positive FCs in heterozygous control (**I**) or homozygous (**J**) of αTub84B^K40A knock-in egg chambers (Stage3-4). **K**) A graph shows frequency of ovarioles containing any Vasa-positive FCs in αTub84B^K40A heterozygous (control) or homozygous (K40A) at days 7, 14, and 21 post-eclosion. Adjusted P values from Šidák's multiple comparisons test are shown for G, H, and K. For all data points, a minimum of 20 ovaries were used from two individual crosses. Scale Bars: 10 μm.

These results support the second scenario in which disturbance of microtubules results in disruption of the junction and plasma membrane between germline cysts and FCs, leading to a leakage of cytoplasmic contents. Because the effect occurs within a short time (30 min), we considered that the treatment is unlikely to allow enough time for the altering FCs' cell-fates, suggesting instead that depolymerizing microtubules may disrupt the barrier between the germline and FC and result in direct leakage of cytoplasmic contents between these cell types. From these results, we suggest that the observed Vasa mislocalization in *lky^KO* is likely caused by direct leakage of cytoplasmic contents between germline and FCs due to a physical disturbance of the boundary between germline cysts and FCs.

## Mislocalization of *nanos (nos)* gene product causes egg chamber fusion in *lky^KO* ovaries

Finally, we wondered if the two observed phenotypes, egg-chamber fusion, and mislocalization (or leakage) of germline contents in FCs, are related each other. We assessed the correlation of the two phenotypes by scoring the co-presence of them at day 14 in *lky^KO* flies, at which both phenotypes are still observed with low frequency (Figs 2G and 3G). We indeed found that these two phenotypes show a strong correlation (S3 Fig). Therefore, we next assessed how mislocalization of germline contents and egg chamber fusion phenotypes are functionally related to each other.

Previously, the tumor suppressor *L(3)mbt* was shown to suppress the expression of the germline-specific gene *nanos (nos)* in FCs. This study demonstrated that misexpression of *nos* in FCs is sufficient to case the egg chamber fusion phenotype [53], similar to what is seen in *lky^KO* ovaries. In addition, the introduction of a hypomorphic allele of *nos* (*nos^L7/nos^BN*) into the *L(3)mbt* mutant background suppressed the egg chamber fusion defect, suggesting that somatic *nos* expression is sufficient to cause egg chamber fusion. As germline contents are transferred to FCs in *lky^KO*, we considered the possibility that those FCs also contain *nos* gene products, leading to egg chamber fusion.

Strikingly, FCs in *lky^KO* ovaries were also positive for expression of a *nosGal4* (Gal4 expressed under the *nos* promoter)-driven fluorescent transgene (*nosGal4>mEOS*). We tested if these FCs also contain *nos* mRNA using fluorescence *in situ* hybridization (FISH) and found that mEOS-positive FCs were also positive for *nos* mRNA (Fig 5A and 5B).

In *lky^KO* ovaries, FCs positive for germline-expressed mEOS contained nos mRNA (Fig 5A and 5B). Therefore, we speculated that nos mRNA leakage into FCs may cause the phenotype of egg chamber fusion. To test whether *nos* in FCs is responsible for the egg chamber fusion in *lky^KO* ovaries, we analyzed ovaries from flies homozygous for both *lky^KO* and hypomorphic alleles of *nos* (*nos^L7/nos^BN*). Strikingly, we observed a significant decrease in the frequency of egg chamber fusion in these flies (Fig 5C). Consistent with the previous report [53], the FC-specific expression of a *nos* transgene caused continuous germline cysts in the germarium,

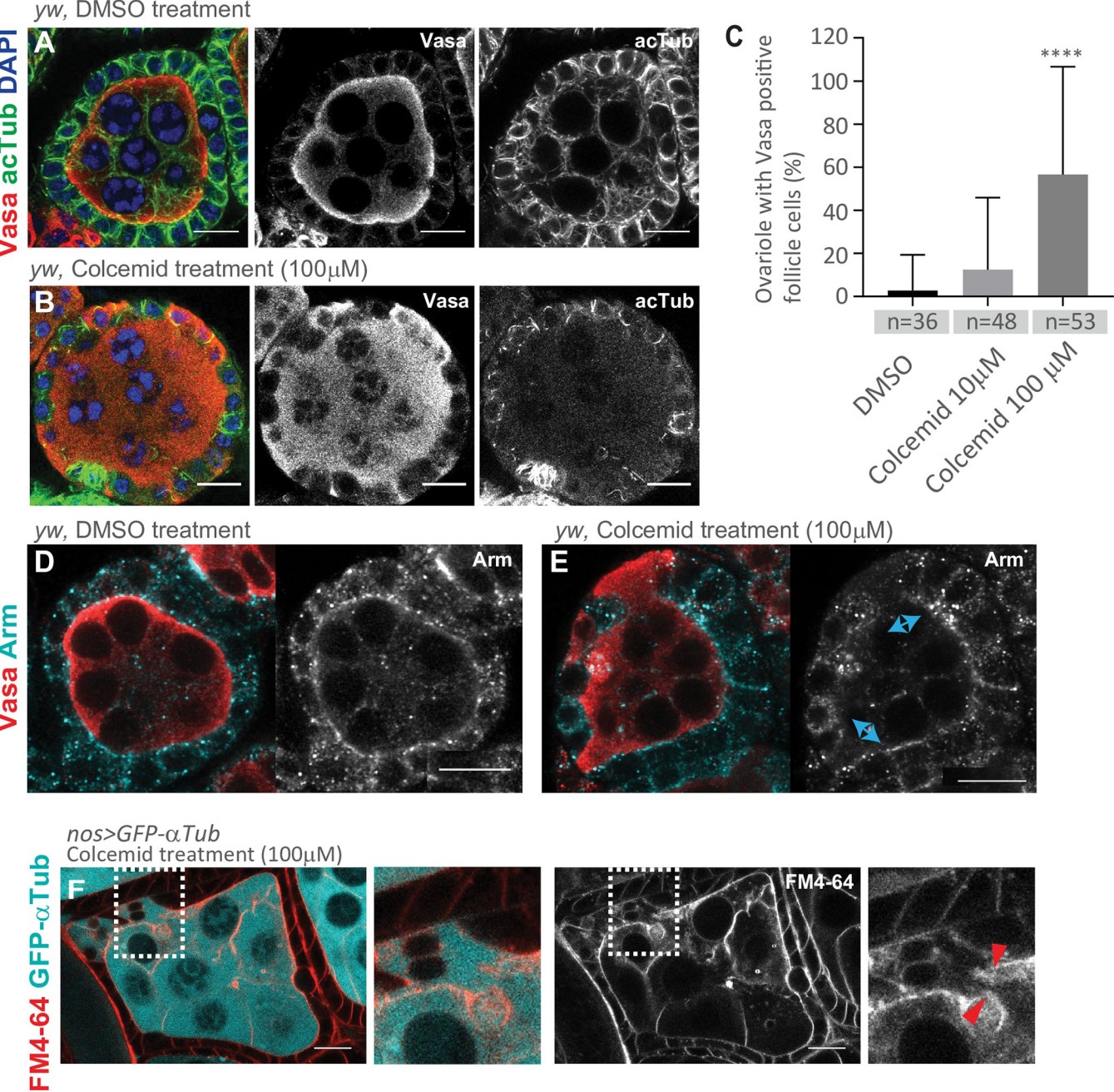

**Fig 4. Short-term Colcemid treatment recapitulates Vasa mislocalization observed in *lky*^KO ovaries. A, B**) Representative images of stage 3–4 egg chambers incubated with DMSO (**A**) or 100 μM Colcemid (**B**) for 30 min after dissection. IF staining was performed for Vasa and acTub. **C**) A graph shows frequency of ovarioles with any Vasa-positive FCs after indicated treatments. **D, E**) Representative images of IF staining for Vasa and Arm in ovaries after incubated with DMSO (control, **D**) or Colcemid (**E**). Blue arrows in **E** indicate germ cell/FC boundary with disrupted junctions visualized by Arm staining. **F**) A representative image of a live egg chamber (stage 3–4) with visualized its plasma membrane by FM4-64 dye after Colcemid treatment. Germline-expressed GFP-αTub shows leakage into FCs through ruptured plasma membrane. Right panels show magnified portion of missing membrane (squared in left panels). For **C**, adjusted P values from Dunnett's multiple comparisons test is shown, and a minimum of 20 ovaries were used from two individual experiments. Scale Bars: 10 μm.

similar to *lky*^KO ovaries (Fig 5C), indicating that the presence of *nos* in FCs is sufficient for these phenotypes. Together, these data suggest that *nos* gene product mislocalizes in somatic FCs in *lky*^KO ovaries, resulting in egg chamber fusion.

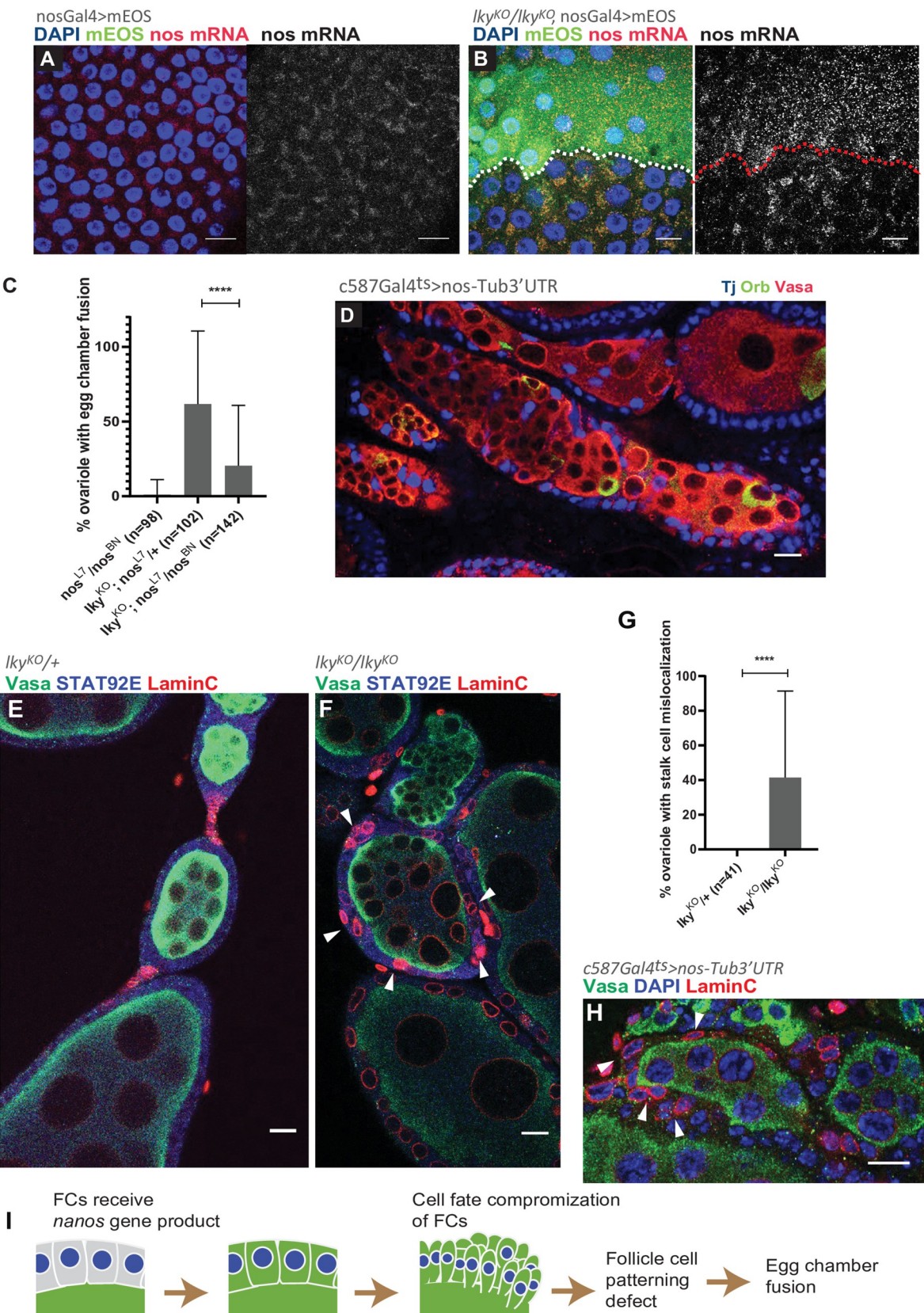

**Fig 5. Mislocalization of *nanos (nos)* gene product causes egg chamber fusion in *lky^KO* ovaries. A-B**) Representative images of *nanos* (*nos*) mRNA FISH of control (**A**) and *lky^KO* (**B**). Surface views of Stage 3 egg chambers are shown. In (**B**), mEOS specifically expressed in the *lky^KO* germline is present in the FCs. Broken lines in **B** indicate the boundary of mEOS-positive (upper side) and negative (lower side) FCs. mEOS-positive FCs are also positive for *nos* mRNA. **C**) A representative image of ectopic expression of *nos* (with *tubulin 3'UTR* to enable translation in somatic cells) using *c587-Gal4; Tub-Gal80^ts* driver shows egg chamber fusion after 4-days temperature shift in 29˚C. **D**) A graph shows percentages of ovarioles with fused egg chambers. Introducing hypomorphic alleles of *nos* (*nos^L7* and *nos^BN*) in the *lky^KO* background reduced the frequency of egg chamber fusion. For **D**, 30 ovaries were used from three individual experiments. **E-F**) IF staining for Vasa, Stat92E, and LaminC, showing stalk cells marked by LaminC and Stat92E expression in control (**E**) versus a fused egg chamber in *lky^KO* (**F**). **G**) A graph shows frequency of ovarioles of indicated genotypes with stalk cell mislocalization. **H**) A representative image of ectopic expression of *nos* using *c587-Gal4; Tub-Gal80^ts* driver showing stalk cell mislocalization. **I**) A model for germline leakage causing egg chamber fusion in *lky^KO* ovaries. Contents from the germline leak into follicle cells and cause a defect in their specification and patterning, leading to egg chamber fusion. Adjusted P value from Dunnett's multiple comparisons test is shown for C, and P value from Student's t-test in G. For all data points, a minimum of 20 ovaries were used from two individual experiments. Scale Bars: 10μm.

It has been reported that the egg chamber fusion is caused by a defect in the specification of stalk cells [54]. In *lky*^KO ovaries (day 21), Lamin-C positive stalk cells were often mislocalized (Fig 5E–5G), suggesting the possibility that transfer of *nos* gene product may affect stalk cell patterning, resulting in the egg chamber fusion phenotype. Consistently, overexpression of *nos* in FCs also showed similar stalk cell mislocalization (Fig 5H).

Based on these data, we propose that loss of acTub in germline causes a leak of germline contents into FCs. FCs that receive *nos* gene product are compromised in their cell fate, which causes aberrant egg chamber development (Fig 5I).

## Discussion

Despite being studied for decades, the physiological role of α-tubulin acetylation is unclear in most cell types. Here we demonstrate that a previously uncharacterized αTAT (α-tubulin transacetylase), Leaky (Lky), is responsible for α-tubulin acetylation in the early stage of germ cells in *Drosophila* ovary. We generated *lky*^KO flies and found two phenotypes; 1) egg-chamber fusion, and 2) mislocalization of germline-specific protein Vasa in neighboring somatic follicle cells. We performed chemical disturbance of microtubules and found that Vasa was quickly transferred from germline to adjacent follicle cells within 30 minutes, suggesting the possibility that the Vasa mislocalization observed in *lky*^KO is likely caused by direct leakage of cellular contents between these two cell types. Therefore, we propose that microtubule acetylation is required for maintaining the cellular identity by preventing the leakage of tissue-specific gene products between neighboring distinct cell types. Finally, we show that the presence of germ-line-specific *nanos* gene product in FC is the cause of the egg chamber fusion.

The molecular role of α-K40-acetylation has been enigmatic because the site of acetylation, K40, is located in the lumen of the microtubules [6, 55], and the acetylated K40 does not seem to be specifically recognized by other microtubule-associating proteins. Therefore, after its discovery in the 1980s [2, 4, 56], it has long been unclear if acetylated α-tubulin K40 is just a passive mark of stable microtubules or if acetylation actively modulates the function of microtubules. Only recently have *in vitro* studies demonstrated that acetylation indeed adds specific function to microtubules, such as enhancing resilience against mechanical stresses, thus potentially contributing to the maintenance of cellular rigidity [9–11].

It is still unknown how Lky-mediated α-K40-acetylation compromises barrier between germline and soma. During the 4 cycles of syncytial divisions, cysts rapidly increase the shared volume, such that a 16-cell cyst may have 16 times more volume after the 4th division. In this process, the fusome branches at the middle portion of cysts and microtubules emanated from the fusome may have an important role in establishment of cellular-rigidity, and α-K40 acetylated microtubule populations may specifically be involved in this process as shown in other systems. When newly formed cysts without proper rigidity experience strong mechanical

forces upon encapsulation by somatic FCs, interacting plasma membrane may rupture, result in abnormal diffusion of cellular contents between neighboring germline and soma. It has been known that small plasma membrane rupture is quickly repaired [57]. We speculate that multiple rupture-repair cycles may occur repeatedly and finally the cell fate is compromised. Further studies will be necessary to determine the underlying mechanism.

Intriguingly, the egg chamber fusion observed in *lky*<sup>KO</sup> flies becomes more prevalent with age. Similarly, reported neuronal defects caused by loss of αTAT often worsen as the animal ages [58]. Therefore, physiological changes in aged animals must play a role in the penetrance of this phenotype. Consistently, when knock-down of *lky* was started at day 0 post-eclosion, the egg chamber fusion phenotype appeared after 14 days or later. However, we observed the phenotype appearing promptly (within 3 days after induction) when knock-down of *lky* was started after flies were aged. Identifying the "aging" factor that affects the penetrance of this phenotype would be a fascinating future study.

In summary, our data demonstrate that α-K40-acetylated microtubules are required in the early germline for proper barrier function between the germline and neighboring FCs. We suggest that this mechanism may be also utilized for maintaining cellular identities in other tissues in broad organisms. Therefore, our findings presented here may contribute to the understanding of multiple previously unrecognized pathological conditions.

## Methods

### Fly husbandry and strains

All fly stocks were raised on standard Bloomington medium at 25˚C. *lky* RNAi flies were placed at 29˚C after eclosion for desired days before dissection. For heat-shock mediated induction of RNAi in aged (day 21) vs. young (day 0–1) flies, *hs-FLP; nos-FRT-stop-FRT-Gal4* [59] was crossed with RNAi lines and progenies were heat-shocked at 37˚C for 60 minutes for 2 times. Then, testes were dissected at the indicated days after the heat shock. Flies cultured for aging were placed into fresh food vials every 48 hours and dissected at indicated ages. Temperature shift was performed by culturing flies at room temperature and shifted to 29˚C upon eclosion for the 4 days before analysis. Combinations of *Tub-Gal80*<sup>ts</sup> (a gift from Cheng-Yu-Lee) with *c587Gal4* (a gift from Yukiko M. Yamashita) were used. The following fly stocks were used:: *αTub84B*<sup>K40A</sup> (FBal0345033, [52] αTub84B knock-in, gift from J. Wildonger), *dTAT*<sup>KO</sup> (FBal0345149, [15], gift from J. Wildonger), *UAS-nosTub3'UTR, nosGal4VP16, tjGal4* (gifts from Yukiko M. Yamashita), *nos*<sup>L7</sup> and *nos*<sup>BN</sup> (gifts from R. Lehmann). The following stocks were obtained from the Bloomington stock center, *w1118* (BDSC 3605), *yw* (BDSC 1495), *lky* deficiency, *Df(1)BSC586* (BDSC 25420) and *UASp-hts.mCherry* (BDSC 66171).

### Generation of transgenic and knock-out lines

**lky knock-out (KO).**   *lky*<sup>KO</sup> flies were generated using the CRISPR/Cas9 system to delete the entire *CG17003* coding region, as described previously [60]. Homology arms were generated using the following primer sets:

EcoRI 5' arm Forward: CTGAATTCATATACCCAGGATGCTAGAGGGTTA
NotI 5' arm Reverse: CTGCGGCCGCGCAGTCGCACTTAGTTCGTTTTCAT
BglII 3' arm Forward: CTAGATCTCTGTAACTTGAGGTCTCGAACTATT
PstI 3' arm Reverse: CTCTGCAGGGAAACTTACAAAAATTTAAGAGGC

Homology arms were then inserted into the pHD-DsRed-attP donor vector ([60], gift from Michael Buszczak) by respective restriction enzyme digest and ligation.

gRNA sequences were as follows:
BbsI 5' gRNA sequence: CTTCGAACTAAGTGCGACTGCATA

BbsI 3' gRNA sequence: `CTTCGAGACCTCAAGTTACAGCCC`

gRNAs were inserted into the pU6-BbsI-gRNA vector ([60], a gift from Michael Buszczak) by BbsI digestion and ligation.

*lky*[KO] CRISPR constructs were injected into Cas9-expressing embryos (nos-Cas9, CAS-0001) by BestGene, Inc. Recombinant flies were crossed with *w1118* (BDSC 3605) and selected by expression of *3xP3-DsRed* (visualized as red eye fluorescence) and validated by genomic PCR (S1 Fig).

Primer sequences for validation are as follows:

5' arm:

`5'-TGTGATTTGCGAATGGGATG-3'`

`5'-CCACCACCTGTTCCTGTA-3'`

3' arm:

`5'-CTTCGAGCCGATTGTTTAG-3'`

`5'-ACACCTTGGAGCCGTACTGGAACT-3'`

***lky* RNAi.**   For construction of lky shRNA plasmid, the following oligonucleotides were used:

Line 1:

`5'-ctagcagtGCGAAATCCTAAACATCATGGtagttatattcaagcataCCATGATGTT TAGGATTTCGCgcg -3';`

`5'- aattcgcGCGAAATCCTAAACATCATGGtatgcttgaatataactaCCATGATGTT TAGGATTTCGCactg -3'`

Line 2:

`5'- ctagcagtGGTAGAGCCCGAGAATTATATtagttatattcaagcataATATAATTC TCGGGCTCTACCgcg -3';`

`5'- aattcgcGGTAGAGCCCGAGAATTATATtatgcttgaatataactaATATAATTCT CGGGCTCTACCactg -3'`

Annealed oligonucleotides were subcloned into NheI and EcoRI sites of pWALIUM20 vector (gift from Yukiko M. Yamashita, [61]). Transgenic flies were generated using strain attP2 by PhiC31 integrase-mediated transgenesis (BestGene).

**UAST-αTubK40Q.**   For construction of UAST-GFP-αTub84B K40Q, site directed mutagenesis was performed using primers:

`5'-GCGGAGGTGATGACTCGTTCAACACCTTC-3',`

`5'-CCACGGTTTGGTCAGACGGCATCTGGCCAT-3'` (K40 site is underlined) from UASP-GFP-αTub84B plasmid (gift from Allan Spradling). Resultant plasmid was used to amplify αTub84B K40Q insert using primers with restriction sites (underlined):

BglII αTub84B-F; `5'-TACAGATCTATGCGTGAATGTATCTCTATCCATG-3',`

NotI αTub84B-R; `5'-GCGGCCGCTTCTGCTATACGTGTCTTTGTGGATAA-3'.`

The amplified fragment was subcloned into BglII/NotI sites of pUAST-GFP-attB vector (gift from Cheng-Yu-Lee) and sequenced. Transgenic flies were generated using strain 24482 by PhiC31 integrase-mediated transgenesis (BestGene).

## UASp-GFP-*lky*

*lky* (CG17003) is an intron-less gene. We amplified cDNA from yw genomic DNA using the following primers with restriction sites (underlined):

BamHI lkyF `5'-TCGGATCCGatggtggagttcgcctttgaca-3'`

AscI lkyR `5'-TAGGCGCGCCttagaatctccggcccccggaaacc-3'`

PCR product was then digested with BamHI and AscI and ligated to a modified pUASP-attB-GFP vector (gift from Michael Buszczak) using BamHI and AscI sites located at the 3' end

of GFP. Transgenic flies were generated using strain attP2 by PhiC31 integrase-mediated transgenesis (BestGene).

## UASp-mEOS

mEOS3.2 fragment was amplified using primers with restriction sites (underlined);

NotI-EOS-F; 5′– TCGCGGCCGCCCCCTTCACCATGagtgcgattaagccagac-3′,

AscI-EOS-R; 5′–ACTGGCGCGCCtcgtctggcattgtcaggcaa-3′ from mEos3.2-C1 vector (gift from Michael Buszczak) and subcloned into NotI/AscI sites of a modified pUASP-attB-GFP vector (gift from Michael Buszczak). Transgenic flies were generated using strain 24749 by PhiC31 integrase-mediated transgenesis (BestGene).

## Immunofluorescent staining

Ovaries were dissected into 1× phosphate-buffered saline (PBS) and fixed in 4% formaldehyde in PBS for 30 minutes, then briefly rinsed three times in PBST (PBS + 0.3% TritonX-100) and permeabilized in PBST for 60min, then incubated in primary antibodies in 3% bovine serum albumin (BSA) in PBST at 4˚C overnight. Samples were then washed three times in PBST for one hour (three 20 minute washes), then incubated in secondary antibodies in 3% BSA in PBST for 2–4 hours at room temperature, or at 4˚C overnight. Samples were then washed three times in PBST for one hour (three 20 minute washes), then mounted using VECTA-SHIELD with 4,6-diamidino-2-phenylindole (DAPI) (Vector Lab, H-1200). Samples were placed on nutator for all incubation steps.

The primary antibodies used were as follows: anti Hts (1B1, 1:20), anti-Orb (1:20), anti rat-Vasa (1:20), anti-LaminC (1:20), and anti-Arm (1:20) were obtained from the Developmental Studies Hybridoma Bank (DSHB), anti-acetylated microtubule antibody (6–11, b-1) was obtained from Sigma (1:200), anti-TJ (1:5000) (gift from Dorothea Godt), anti-Stat92E (1:2000, gift from Yukiko M. Yamashita), and anti-rabbit Vasa (1:200) (Santa Cruz Biotechnology).

AlexaFluor-conjugated secondary antibodies were used at a dilution of 1:400. Images were taken using a Zeiss LSM800 with Airyscan with a 63 ×oil immersion objective (NA = 1.4) and processed using Fiji. For comparisons of two or more samples, identical parameters were used for image acquisition and processing across samples.

## Fluorescent *in situ* hybridization

Fluorescent in situ hybridization was performed as described previously [62] with some modi-fications. Briefly, ovaries were dissected in 1X PBS and then fixed in 4% formaldehyde/PBS for 45 minutes at room temperature. After fixation, ovaries were briefly rinsed 2 times with 1X PBS, then resuspended in ice-cold 70% EtOH, and incubated at 4˚C for 2 hours. Then, ovaries were rinsed briefly in wash buffer (2X SSC and 10% deionized formamide), then hybridized for 16 hours at 37˚C in the dark with 50 nM of Quasar 570 labeled Stellaris probe against entire *nanos* 3′UTR sequence or *lky* full length cDNA sequence (LGC Biosearch Technologies) in the Hybridization Buffer containing 2X SSC, 10% dextran sulfate (MilliporeSigma), 1 μg/μl of yeast tRNA (MilliporeSigma), 2 mM vanadyl ribonucleoside complex (NEB), 0.02% RNase-free BSA (ThermoFisher), and 10% deionized formamide. Then ovaries were washed 2 times for 30 minutes each at 37˚C in the dark in the prewarmed wash buffer (2X SSC, 10% formam-ide) and then resuspended in a drop of VECTASHIELD with DAPI. Samples were placed on nutator for all incubation steps. Images were taken using a Zeiss LSM800 with Airyscan with a 63 ×oil immersion objective (NA = 1.4) and processed using Fiji.

## Quantitative RT-PCR

Germaria from flies of desired age were hand dissected and collected and homogenized by pipetting in TRIzol Reagent (ThermoFisher), and RNA was extracted following the manufacturer's instructions. RNA was then reverse transcribed to cDNA using SuperScript III First-Strand Synthesis Super Mix (ThermoFisher) with Oligo (dT)20 Primer. Quantitative PCR was performed, in duplicate, using SYBR green Applied Biosystems Gene Expression Master Mix on a CFX96 Real-Time PCR Detection System (Bio-Rad). A control primers for act5C (5′–GCCAGCAGTCGTCTAATCCA–3′/5′–GCATCGTCTCCGGCAAAT –3′) and primers for *lky* (5′–TTCGACGATCCCATCACCAC–3′/5′–GCCCATGTTTTATCTGCGCC–3′) were used. Relative quantification was performed using the comparative CT method (ABI manual).

## Colcemid treatment

Ovaries were dissected into 1 ml of pre-warmed Schneider's *Drosophila* medium supplemented with 10% fetal bovine serum and glutamine–penicillin–streptomycin. Colcemid (MilliporeSigma, dissolved in DMSO) was added to a final concentration of either 10 or 100 μM. Control samples were incubated with identical amount of DMSO. The ovaries were incubated at room temperature (22–23°C) for 30 min, then washed twice with PBS before fixation in 4% paraformaldehyde for 30 minutes. Ovaries were then processed for Immunofluorescent staining as described above.

## Live imaging

For live observation of unfixed ovaries, ovaries were dissected in 1 ml of pre-warmed Schneider's *Drosophila* medium supplemented with 10% fetal bovine serum and glutamine–penicillin–streptomycin. Colcemid (100 μM) were added to the media and incubated at room temperature (22–23°C) for 30 min. Ovaries were then placed on microscope slide with coverslip spacers on both edges, and another coverslip was placed on top. Hoechst 33342 (2 μg/ml) and FM4-64FX Lipophilic Styryl Dye (5 μg/ml, Molecular Probes) were added 1 min before analysis. Imaging was performed in the presence of dye within 15 min, using a Zeiss LSM800 with Airyscan with a 63 ×oil immersion objective (NA = 1.4) and processed using Fiji.

## Statistical analysis and graphing

The "egg chamber fusion" phenotype was determined as egg chambers containing more than 16 germ cells encapsulated by a single layer of follicle cells, as seen by immunofluorescence staining with Vasa and DAPI.

The "vasa positive follicle cell" phenotype was determined by the presence of Vasa in one or more follicle cells in an egg chamber, as seen by immunofluorescence staining with Vasa, and DAPI and/or Traffic jam.

To quantify the frequency of phenotypes, ovarioles were scored using a YES/NO criteria. Ovarioles from a minimum of ten ovaries were dissected from flies in the same culture and scored as containing one or more egg chambers exhibiting the phenotype of interest ("YES"), or no egg chambers exhibiting the phenotype of interest ("NO"), and the total number of ovarioles with the phenotype was calculated as a frequency (percentage) based on the results of at least two independent experiments.

No statistical methods were used to predetermine sample size. The experiment values were not randomized. The investigators were not blinded to allocation during experiments and outcome assessment. Statistical analysis and graphing were performed using GraphPad prism9 or Microsoft excel. Individual numerical values displayed in all graphs are provided in S1 Data.

All data are shown as means ± standard deviation (s.d.) The P values from Student's t-test or adjusted P values from Dunnett's multiple comparisons or Šidák's multiple comparisons test are provided; shown as $^*$P<0.05, $^{**}$P<0.01, $^{***}$P<0.001, $^{****}$P<0.0001; NS, non-significant (P≥0.05). Details for each test used are reported in the figure legends for each graph.

## Supporting information

**S1 Fig.** **A**) A schematic of the construct used to generate $lky^{KO}$ flies. **B**) Validation of the KO genotype by PCR. **C-F**) IF staining for acTub and Vasa of germaria of the indicated genotypes. "double KO" = $lky^{KO}/lky^{KO};; dTAT^{KO}/dTAT^{KO}$. **G**) A representative image of a tip of an ovariole of wild type control (*yw*) and *Double KO* showing remaining acTub in later stages of germline and FCs. Flies were used at 0–7 days post-eclosion. Scale Bars; 10μm. Flies were used at 0–7 days post-eclosion. Scale Bars; 10μm.
(TIF)

**S2 Fig.** **A**) A graph shows level of *lky* mRNA relative to *act5C* mRNA (lky/act5C) in germaria isolated from indicated ages. Wildtype (*yw*) flies were used (See *Methods* for details). **B-C**) Representative images of germaria of IF staining for acTub (cyan) and Vasa (red). *lky* RNAi was induced by heat shock (HS) before (1 day post-eclosion) or after aging animals (21 days post-eclosion). **D**) A graph shows frequency of ovarioles with egg chamber fusion after heat shock at the indicated ages. Adjusted P values from Šidák's multiple comparisons test are shown. For all data points, a minimum of 20 ovaries were used. Scale bars; 10μm.
(TIF)

**S3 Fig.** A) Graph shows comparison of frequency of ovarioles with Vasa positive FCs within fusion positive vs. fusion negative ovarioles. B) Graph shows comparison of frequency of ovarioles with fusion within ovarioles with or without Vasa positive FCs. Flies were used at day14 post-eclosion. P values from Student's t-test are shown. For all data points, a minimum of 20 ovaries were used.
(TIF)

**S1 Data. Individual numerical values displayed in all graphs are provided.**
(XLSX)

**S1 Raw images. Original images for gels are provided.**
(TIF)

## Acknowledgments

We thank Yukiko M. Yamashita, Michael Buszczak and Marie Bao for reading our manuscript and valuable discussions. Yukiko M. Yamashita, Michael Buszczak, Jill Wildonger, Jay Z Parrish, Dorothea Godt, Cheng-Yu-Lee, Allan C. Spradling, Ruth Lehmann and the Bloomington *Drosophila* Stock Center and the Developmental Studies Hybridoma Bank for reagents. Christopher Bonin for manuscript editing.

## Author Contributions

**Conceptualization:** Matthew Antel, Taylor Simao, Mayu Inaba.

**Funding acquisition:** Mayu Inaba.

**Investigation:** Matthew Antel, Taylor Simao, Muhammed Burak Bener, Mayu Inaba.

**Project administration:** Mayu Inaba.

**Writing – original draft:** Matthew Antel.

**Writing – review & editing:** Matthew Antel, Taylor Simao, Muhammed Burak Bener, Mayu Inaba.

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
