## [Decision Letter · Decision Letter 0]

22 Aug 2022

PONE-D-22-15901

Supplementary information for Drosophila CG17003/leaky (lky) is required for microtubule acetylation in early germ cells in Drosophila ovary

PLOS ONE

Dear Dr. Inaba,

Thank you for submitting your manuscript to PLOS ONE. After careful consideration, we feel that it has merit but does not fully meet PLOS ONE’s publication criteria as it currently stands. Therefore, we invite you to submit a revised version of the manuscript that addresses the points raised during the review process.

Specifically you'll see that one of the reviewer is concerned by the statistical analyzes. Please provide answers to each of his comments

We look forward to receiving your revised manuscript.

Kind regards,

Claude Prigent

Academic Editor

PLOS ONE

Journal Requirements:

   "This research is supported by R35GM128678 from National Institute for General Medical Sciences and start-up funds from UConn Health (to M.I.)."

   "We thank Yukiko M. Yamashita, Michael Buszczak and Marie Bao for reading our manuscript and valuable discussions. Yukiko M. Yamashita, Michael Buszczak, Jill Wildonger, Jay Z Parrish, Dorothea Godt, Cheng-Yu-Lee, Allan C. Spradling, Ruth Lehmann and the Bloomington Drosophila Stock Center and the Developmental Studies Hybridoma Bank for reagents. Christopher Bonin for manuscript editing. This research is supported by R35GM128678 from National Institute for General Medical Sciences and start-up funds from UConn Health (to M.I.)."

   "This research is supported by R35GM128678 from National Institute for General Medical Sciences and start-up funds from UConn Health (to M.I.)."

Reviewers' comments:

Reviewer's Responses to Questions

**Comments to the Author**

1. Is the manuscript technically sound, and do the data support the conclusions?

Reviewer #1: Yes

Reviewer #2: Yes

2. Has the statistical analysis been performed appropriately and rigorously? 

Reviewer #1: Yes

Reviewer #2: I Don't Know

3. Have the authors made all data underlying the findings in their manuscript fully available?

Reviewer #1: Yes

Reviewer #2: Yes

4. Is the manuscript presented in an intelligible fashion and written in standard English?

Reviewer #1: Yes

Reviewer #2: Yes

5. Review Comments to the Author

Reviewer #1: This paper represents another step in an unfolding story of how

Microtubule (MT) acetylation function in a developmental context. Among their results, the authors demonstrate that both manipulation of the CG17003/leaky (lky), uncharacterized αTAT, and also generating a non- acetylatable α-tubulin84BK40A Drosophila strain, led to age dependent defects in oogenesis. Manipulation also of MT by drug treatment, led to similar phenotype as loss of lky, namely progressive egg chamber fusion phenotype accompanied with mislocalization of germline-specific Vasa protein in somatic follicle cells. The Authors also showed that probably the "leakage" or mis-localization of Vasa to the follicle cells, may be the cause for progressive egg chamber fusion phenotype, since mutations in nanos suppressed the egg chamber fusion defect (similar results were shown before for the tumor suppressor L(3)mbt , where they showed that L(3)mbt suppressed the expression of germline genes in the follicle which led to egg chamber fusion defect. Still, the molecular mechanism by which MT acetylation affecting the "leakage" or mis-localization of Vasa to the follicle cells, is still to be revealed. I recommend it for publication as is.

Reviewer #2: The manuscript presents roles of a tubulin acetyl transferase, CG17003/leaky, in early Drosophila oogenesis. Knockout of leaky resulted in loss of K40 acetylation in alpha-tubulin in early germline cells during oogenesis, and an age-dependent egg chamber fusion which was phenocopied by expression of a non-acetylatable alpha-tubulin mutant. Furthermore, Leaky knockout led to loss of cell membrane integrity and leakage of cellular contents from germline cells to follicle cells. This is an interesting piece of work, reporting novel findings.

My issues are mainly related to the statistical analysis. It is difficult to judge whether the data supports their conclusions, as it is not clearly described how the data was statistically analysed.

(1) Many conclusions rely on differences at p<0.05 (*), which seems weak. At least for key conclusions, it would be vital to increase the sample numbers to allow firmer conclusions.

(2) n indicated in some figures seems the total number of egg chambers examined, but to calculate both % and SD, they must have been divided into smaller batches (for example, egg chambers from one animal). This procedure should be described either in figure legends or materials & methods.

(3) "The P values from Student’s t-test or adjusted P values from Dunnett’s multiple comparisons test are provided." It is not clear which data was analysed by which method.

(4) I am not sure about some p-values, although it is difficult to judge the validity without knowing the precise methodology. For example, in Figure 2H, 43 ± 9% (mean ± s.d.) compared with 10 ± 1% should give a lower p value than 0.01 (even if n=3) in t-test. Do the error bars really represent SDs rather than SEMs? Or has any adjustment been made for the p-value?

(5) Fig1I. t-test does not look appropriate in this case, as this test assumes normal distributions (all have "outliers" except lky KO GM). Another method that does not assume normal distribution, such as Mann Whitney U test, would be more appropriate.

(6) It may be a good practice to include spreadsheets containing raw numbers used in statistical analysis with actual p-values as supporting information.

Also the following issues should be addressed.

(7) Inclusion of page numbers in the manuscript would be helpful to give comments.

(8) Need description about how images are captured and processed (including contrast changes) for figures. This is especially crucial when any statements are made about differences in the intensity between control and mutants (whether they are equally captured and processed).

(9) Fig S1G need a wild-type control image.

6. PLOS authors have the option to publish the peer review history of their article (what does this mean?). If published, this will include your full peer review and any attached files.

Reviewer #1: No

Reviewer #2: No

---

## [Author Response · Author response to Decision Letter 0]

17 Sep 2022

First of all, we thank editor and the reviewers for pointing out critical issues on our original manuscript. Thankfully, the revised version has been improved significantly strengthened our conclusions.

Reviewer #1

This paper represents another step in an unfolding story of how

Microtubule (MT) acetylation function in a developmental context. Among their results, the authors demonstrate that both manipulation of the CG17003/leaky (lky), uncharacterized αTAT, and also generating a non- acetylatable α-tubulin84BK40A Drosophila strain, led to age dependent defects in oogenesis. Manipulation also of MT by drug treatment, led to similar phenotype as loss of lky, namely progressive egg chamber fusion phenotype accompanied with mislocalization of germline-specific Vasa protein in somatic follicle cells. The Authors also showed that probably the "leakage" or mis-localization of Vasa to the follicle cells, may be the cause for progressive egg chamber fusion phenotype, since mutations in nanos suppressed the egg chamber fusion defect (similar results were shown before for the tumor suppressor L(3)mbt , where they showed that L(3)mbt suppressed the expression of germline genes in the follicle which led to egg chamber fusion defect. Still, the molecular mechanism by which MT acetylation affecting the "leakage" or mis-localization of Vasa to the follicle cells, is still to be revealed. I recommend it for publication as is.

We appreciate this reviewer for clearly summarizing our work and his/her positive recommendation. 

Reviewer #2

The manuscript presents roles of a tubulin acetyl transferase, CG17003/leaky, in early Drosophila oogenesis. Knockout of leaky resulted in loss of K40 acetylation in alpha-tubulin in early germline cells during oogenesis, and an age-dependent egg chamber fusion which was phenocopied by expression of a non-acetylatable alpha-tubulin mutant. Furthermore, Leaky knockout led to loss of cell membrane integrity and leakage of cellular contents from germline cells to follicle cells. This is an interesting piece of work, reporting novel findings.My issues are mainly related to the statistical analysis. It is difficult to judge whether the data supports their conclusions, as it is not clearly described how the data was statistically analysed.

We thank the reviewer for pointing out issues on our original analysis. We agree that the statistical analyses performed in our original manuscript were not appropriate, which were calculated among a grouped analysis. In the revised version, we reanalyzed all the data so that the SD reflects variance of each sample. All figures have been updated with new p-values and SDs, and a description of the statistical tests used is included in each figure legend. 

(1) Many conclusions rely on differences at p<0.05 (*), which seems weak. At least for key conclusions, it would be vital to increase the sample numbers to allow firmer conclusions. 

Based on this new analysis, we obtained overall lower p-values and we believe the original data is meaningful without the need to increase sample numbers. 

(2) n indicated in some figures seems the total number of egg chambers examined, but to calculate both % and SD, they must have been divided into smaller batches (for example, egg chambers from one animal). This procedure should be described either in figure legends or materials & methods.

We apologize again for this confusion. As this reviewer says, we originally did grouped analysis using percentage from each biological replicate, it made our overall p values very high and the “n” should have been number of biological replication (n=2 or 3).

As stated in above response, we have re-analyzed all of the original data using the appropriate tests (such that single ovariole does have fusion or leak, we input 1. If it does not have fusion or leak, we put zero. In this way, now “n” reflects analyzed sample number). We reanalyzed all the data and all figures have been updated with new p-values, SDs, and a description of the statistical tests used is included in each figure legend. 

(3) "The P values from Student’s t-test or adjusted P values from Dunnett’s multiple comparisons test are provided." It is not clear which data was analysed by which method.

We have added details of analyses in all figure legends.

(4) I am not sure about some p-values, although it is difficult to judge the validity without knowing the precise methodology. For example, in Figure 2H, 43 ± 9% (mean ± s.d.) compared with 10 ± 1% should give a lower p value than 0.01 (even if n=3) in t-test. Do the error bars really represent SDs rather than SEMs? Or has any adjustment been made for the p-value?

We used SD for all data. But what we did originally was grouped analysis using percentage from each biological repeat (this case, statistically n=2), it made our overall p values very high (See our response above for comment#2). We reanalyzed this data along with all others.

(5) Fig1I. t-test does not look appropriate in this case, as this test assumes normal distributions (all have "outliers" except lky KO GM). Another method that does not assume normal distribution, such as Mann Whitney U test, would be more appropriate.

We thank the reviewer for bringing this to our attention. We agree that those data sets contain outliers and have performed Mann Whitney U test, finding the previously reported difference between lkyKO GM and Control GM is no longer significant. We performed staining again and took high resolution image of germline cysts. We used neighboring follicle cell layer as internal control and calculated relative intensities for each data point. We found significant difference between lkyKO vs control in germline cysts up to stage 5, and have edited the text and Figure 1H-I to reflect this.

(6) It may be a good practice to include spreadsheets containing raw numbers used in statistical analysis with actual p-values as supporting information.

We have created a spreadsheet containing all of the raw data used in the figures along with the actual p-values reported.

Also the following issues should be addressed.

(7) Inclusion of page numbers in the manuscript would be helpful to give comments.

We apologize that we forgot to include page numbers in our original manuscript. We have done so in the revised version.

(8) Need description about how images are captured and processed (including contrast changes) for figures. This is especially crucial when any statements are made about differences in the intensity between control and mutants (whether they are equally captured and processed).

We have added a description of image processing in the Methods section. In summary, for a direct comparison between samples, the samples were stained at the same time using the same methodology, and imaged with the same settings. 

(9) Fig S1G need a wild-type control image.

Thank you, we have included a wild-type image for this figure.

---

## [Decision Letter · Decision Letter 1]

12 Oct 2022

Drosophila CG17003/leaky (lky) is required for microtubule acetylation in early germ cells in Drosophila ovary

PONE-D-22-15901R1

Dear Dr. Inaba,

We’re pleased to inform you that your manuscript has been judged scientifically suitable for publication and will be formally accepted for publication once it meets all outstanding technical requirements.

Kind regards,

Claude Prigent

Academic Editor

PLOS ONE

Additional Editor Comments (optional):

Reviewers' comments:

Reviewer's Responses to Questions

**Comments to the Author**

1. If the authors have adequately addressed your comments raised in a previous round of review and you feel that this manuscript is now acceptable for publication, you may indicate that here to bypass the “Comments to the Author” section, enter your conflict of interest statement in the “Confidential to Editor” section, and submit your "Accept" recommendation.

Reviewer #2: All comments have been addressed

2. Is the manuscript technically sound, and do the data support the conclusions?

Reviewer #2: Yes

3. Has the statistical analysis been performed appropriately and rigorously? 

Reviewer #2: Yes

4. Have the authors made all data underlying the findings in their manuscript fully available?

Reviewer #2: Yes

5. Is the manuscript presented in an intelligible fashion and written in standard English?

Reviewer #2: Yes

6. Review Comments to the Author

Reviewer #2: My issues with the original manuscript were mainly related to the statistical analysis. This revised version clearly states which statistical method was used for each analysis. In addition, the use of more suitable and sensitive statistical methods has helped to obtain more convincing p-values. One remaining issue is that I could not find what the error bars represent in the graphs. Provided that this information is included in the figure legends, I fully support the publication of the manuscript in PLoS ONE.

7. PLOS authors have the option to publish the peer review history of their article (what does this mean?). If published, this will include your full peer review and any attached files.

Reviewer #2: No

---

## [Editor Report · Acceptance letter]

28 Oct 2022

PONE-D-22-15901R1 

*Drosophila CG17003/leaky* (*lky*) is required for microtubule acetylation in early germ cells in *Drosophila* ovary 

Dear Dr. Inaba:

I'm pleased to inform you that your manuscript has been deemed suitable for publication in PLOS ONE. Congratulations! Your manuscript is now with our production department. 

Kind regards, 

on behalf of

Dr. Claude Prigent 

Academic Editor

PLOS ONE